# Electrophysiology as a Tool to Decipher the Network Mechanism of Visceral Pain in Functional Gastrointestinal Disorders

**DOI:** 10.3390/diagnostics13040627

**Published:** 2023-02-08

**Authors:** Md Jahangir Alam, Jiande D. Z. Chen

**Affiliations:** Division of Gastroenterology and Hepatology, Department of Internal Medicine, University of Michigan, Ann Arbor, MI 48109, USA

**Keywords:** visceral pain, functional gastrointestinal disorders, extracellular recordings, neural oscillations, local field potentials

## Abstract

Abdominal pain, including visceral pain, is prevalent in functional gastrointestinal (GI) disorders (FGIDs), affecting the overall quality of a patient’s life. Neural circuits in the brain encode, store, and transfer pain information across brain regions. Ascending pain signals actively shape brain dynamics; in turn, the descending system responds to the pain through neuronal inhibition. Pain processing mechanisms in patients are currently mainly studied with neuroimaging techniques; however, these techniques have a relatively poor temporal resolution. A high temporal resolution method is warranted to decode the dynamics of the pain processing mechanisms. Here, we reviewed crucial brain regions that exhibited pain-modulatory effects in an ascending and descending manner. Moreover, we discussed a uniquely well-suited method, namely extracellular electrophysiology, that captures natural language from the brain with high spatiotemporal resolution. This approach allows parallel recording of large populations of neurons in interconnected brain areas and permits the monitoring of neuronal firing patterns and comparative characterization of the brain oscillations. In addition, we discussed the contribution of these oscillations to pain states. In summary, using innovative, state-of-the-art methods, the large-scale recordings of multiple neurons will guide us to better understanding of pain mechanisms in FGIDs.

## 1. Introduction

Pain is an unpleasant experience containing sensory and emotional components. The unpleasant percept that dominates the affective dimension of pain is coupled with the motivational drive and guides us to escape from potentially harmful events associated with actual or potential tissue damage. The experience of pain is evoked by noxious stimuli or through adverse emotional events and memories. The brain receives nociceptive signals and integrates sensory, cognitive, and emotional information to respond appropriately. Pain has two distinct types, acute and chronic pain, and is very common in patients with functional gastrointestinal (GI) disorders (FGIDs). FGIDs are a group of GI conditions characterized by abdominal pain or discomfort, stool irregularities, bloating, and other somatic, visceral, and psychiatric comorbidities. FGIDs account for 40–50% of all new referrals to GI clinics [1,2]. Despite extensive research in humans and animals, the pathophysiology of FGIDs is still incompletely understood. Visceral pain and pain-related symptoms in FGIDs contribute to substantial psychological distress, functional disability, and healthcare costs. The most common FGIDs are irritable bowel syndrome (IBS)—characterized by recurrent abdominal pain associated with altered bowel habits, psychiatric conditions such as depression and anxiety, and visceral sensitivity—and Functional Dyspepsia (FD)—characterized by chronic indigestion, abdominal pain, and a feeling of fullness or bloating during and after meals.

FGIDs are considered a bio-psycho-social disorder of gut-brain interaction with unclear etiology. Despite differences in etiology, gut-brain communication is essential for understanding the pathophysiology of visceral pain in FGID [3,4,5]. However, whether brain abnormalities drive the gut symptoms or whether changes in the gut alter brain function remains poorly understood. Bidirectional connections between the gut and the central nervous system (CNS) contribute to many neurobiological functions, including microbial, immunological, metabolic, hormonal, and neural processes [6,7]. The brain–gut axis consists of a hierarchy of reflex loops assuring homeostatic control of GI function, and dysregulation of these reflexes can lead to chronic visceral pain in FGID patients [8].

In the clinical setting, pain (visceral pain) processing is mainly studied using hemodynamic brain imaging techniques such as functional magnetic resonance imaging (fMRI) and positron emission tomography (PET), and neuro-electrical techniques, including electroencephalography (EEG), and magnetoencephalography (MEG). These studies have shown that changes in brain structure and function are correlated with persistent pain. In IBS patients, the brain regions associated with attention, including the anterior cingulate cortex (ACC), medial PFC (mPFC), and thalamus, are more active than healthy controls and may underlie symptom-related anxiety and visceral hypersensitivity [9]. Both male and female patients with IBS were shown to have upregulation of the emotional arousal circuit and altered serotonergic modulation of this circuitry was reported to play a role in centrally mediated visceral hypersensitivity [9,10]. These findings suggest modified descending pain modulation systems in patients with FGIDs. Moreover, increased excitability of the spinal dorsal horn (SDH) results in increased ascending input to brain regions processing interoceptive information [11]. IBS patients also demonstrated alteration in grey matter density in brain areas associated with cognitive and evaluative functions [12,13,14,15]. Another study hypothesized that these grey matter abnormalities might be linked to white matter changes [16]. As expected, neuroimaging studies in female IBS patients revealed white matter abnormalities in pain-related brain areas, including the ACC and IC, suggesting that these white matter changes drive the functional and grey matter abnormalities in FGID patients. In addition, neuroimaging studies have reported that FD patients show abnormal brain activities in response to gastric distension, and another study also suggested abnormalities in white matter microstructure [17,18]. Taken together from the studies discussed here, it can be summarized that visceral stimuli represent abnormal brain activity in FGID patients suggesting the alteration of ascending and descending nociceptive pathways, which may underlie the pathophysiology of visceral pain in FGIDs.

## 2. Animal Models in Visceral Pain Study

Rats and mice are the most commonly used animal models for studying visceral hypersensitivity (VH) in FGIDs. Multiple rodent models of visceral pain have been described. However, a few have been well-established, including intracolonic administration of different active compounds or experimental infection with biological agents. Here we briefly discuss some popular methods used to induce VH in rats and mice.

### 2.1. Chemical Stimulation-Induced Animal Models

Infusing chemical substances such as acetic acid (AA) into the descending colon and rectum is commonly used to establish VH. The VH in adulthood can be induced by intrarectal infusion of 0.2 mL of 0.5% AA solution in saline in 10 days old pups; the animals develop VH in response to colorectal distension (CRD) in adulthood [19]. Adult rats that underwent colonic installation of a low volume of mustard oil enema neonatally exhibited chronic VH [20]. Intracolonic injection of zymosan suspension induces VH in rats (25 mg/mL; 1 mL) [21] and mice (30 mg/mL, 0.1 mL, daily for 3 days) [22]. The visceromotor response to CRD can be measured either acutely (2–3 h post-infusion) or following a once-daily infusion for 3 days. Dextran sodium sulfate (DSS) is another chemical that induces mucosal colitis in rats or mice when administered as a 1–5% solution with drinking water for 5–10 days [23]. Another model includes the intrarectal injection of 2,4,6-trinitrobenzene sulfonic acid (TNBS) in 25–50% ethanol, producing acute colonic inflammation and disrupting the mucosal barrier in rats and mice [24,25]. These inflammatory changes in the colonic tissue can be assessed within hours of the infusion, and visceral hypersensitivity in response to distension can be measured 3–7 days post-infusion.

### 2.2. Corticotropin-Releasing Factor (CRF) Induced Animal Models

Corticotropin-releasing factor (CRF) is a crucial regulator of the hypothalamic–pituitary–adrenal (HPA) axis. It is involved in stress response in the brain–gut axis and is considered an essential mediator of visceral hypersensitivity. Previous studies reported that microinjection of CRF into the cerebral ventricle or the central nucleus of the amygdala (CeA) with a dose of 0.1–10 μgkg^−1^ increased colonic motility, visceral perception, and induced anxiety-like behavior [26,27]. These effects can be blocked by administrating CRF receptor type 1 (CRF-1) antagonist CP-15426 [27] or JTC017 [28]. Moreover, CRF-1 knockout mice have shown decreased colonic sensitivity in response to CRD [29], suggesting the importance of CRF in colonic sensitivity.

### 2.3. Neonatal Maternal Separation-Induced Models

Stress in early life, including childhood negligence, and physical or sexual abuse, could have detrimental impacts on individuals. Previous evidence suggests that early life stressors like neonatal maternal separation increase the risk of developing IBS in adulthood [30,31]. The most commonly used model involves the separation of neonatal rats from their mother and nest for 3 h daily between postnatal days (PN) 2–14 [32], mimicking the early deprivation of maternal care. These animals exhibited an increased visceromotor response (VMR) to CRD, suggesting that maternal separation caused visceral hypersensitivity in adulthood [32].

### 2.4. Water Avoidance Stress-Induced Animal Model

Water avoidance stress (WAS) is a non-invasive technique to induce visceral hypersensitivity in rodents. The most conventional approach to constructing this experimental model is to place animals on a dry platform surrounded by water in a plexiglass tank. The animal is placed on the platform for only 1 h (acute stressor) or 1 h daily for 10 consecutive days (chronic stressor). Some studies demonstrated that a single exposure to WAS induced visceral hypersensitivity [33], while other studies have reported that 10 days of consecutive WAS can increase visceral sensitivity in animals [34]. Brain imaging studies demonstrated that rats exposed to chronic WAS had an altered pattern of brain activation similar to that observed in IBS patients [34].

### 2.5. Biological Agents Induced Animal Model

Chronic and persistent IBS symptoms could develop after the remission of acute intestinal infection, known as post-infectious irritable bowel syndrome. Biological agents such as bacteria or parasites can destroy the balance of intestinal microecology and effectively induce visceral hypersensitivity in rats and mice. Bacterial infection with *Citrobacter rodentium* (*C. Rodentium*) caused transient visceral hypersensitivity in mice and persisted for about 2–3 weeks [35]. *Trichinella spiralis* (*T. spiralis*) infection is commonly used to produce long-term colonic hypersensitivity in rats and mice. In response to CRD, these animals demonstrated increased EMG and AWR scores in rats [36] and mice [37], respectively.

### 2.6. Animal Models of Visceral Hypersensitivity Relevant to Functional Dyspepsia

Several rat models of FD have been mentioned previously; however, we will discuss a few of the most popular animal models. Oral gavage of 0.2 mL of 0.1% iodoacetamide in 2% sucrose to male neonatal pups on PN10-16 can induce FD-like responses to gastric balloon distension in adulthood [38]. Administering 0.2 mL of TNBS (130 mg/kg in 10% EtOH in saline) through a tube inserted 2 cm into the distal colon to neonatal rats on PN10 resulted in increased gastric sensitivity in response to stomach distension in adulthood [39]. Another model includes AA (20% AA in sterile saline) injection into the submucosal layer of the gastric wall of adult male rats [40]. However, this method requires surgery and injection at 15–20 sites (10 μL/site).

Although it is challenging to completely mimic visceral pain in animal models, these models provide an opportunity to investigate the visceral pain mechanism in conjunction with the brain–gut axis.

It is worth comparing the differences among different animal models; however, the data is limited to the best of our knowledge as all animal models share the standard mechanisms for inducing VH, although some differences still exist. Herein, we very concisely summarized these observations in Table 1.

## 3. Brain Regions Involved in Pain Processing

Studies ranging from humans to animals, including neuroimaging, functional, anatomical, and biochemical, suggest the involvement of different brain nuclei in pain processing associated with FGIDs. This section will discuss key brain regions involved in nociception.

### 3.1. Anterior Cingulate Cortex (ACC)

To better understand pain perception and develop novel therapies for chronic pain disorders, it is important to exploit the neuronal circuitry associated with pain processing. Research from human and animal studies has emerged to show that the anterior cingulate cortex (ACC) is a critical brain region necessary for processing the emotional aspects of pain [49,50]. The ACC (Figure 1) is a limbic structure activated in acute and chronic pain. The ACC neurons receive inputs from various cortical and subcortical structures, including the thalamus [50,51,52], the amygdala, and the insular cortex (IC). These inputs convey nociceptive information from somatic and visceral organs to the ACC. Electrophysiological recordings from ACC neurons in humans and experimental animals demonstrated that ACC neurons respond to noxious stimuli and show increased responses to more significant pain intensity. Nociceptive-specific ACC neurons responded to painful somatic thermal and mechanical stimuli; while some neurons had restricted receptive fields, others had more complex responses possibly related to higher integrative or cognitive functions [53]. Similar results have been observed in nonhuman primates and rodent studies [53,54,55,56]. Animal behavioral studies found that inhibition of ACC activity through chemical or electrolytic lesions attenuates the affective component of the pain state [57,58], suggesting the role of ACC in pain perception. However, in the absence of peripheral noxious stimulation, chemical stimulation of the ACC is sufficient to induce pain behavior [57] in rodents. With recent advances in methodological approaches, optogenetics has been used extensively to manipulate specific cell types and circuits. Excitation of ACC pyramidal neurons and activation of the medial thalamus-ACC pathway lowers mechanical pain thresholds in mice [59] and conveys aversive information during chronic pain [60], respectively. By combining optogenetics with fMRI (ofMRI), a recent study in mice revealed that the ACC and its downstream neural circuit are vulnerable to chronic pain. The ACC is also involved in maintaining chronic pain hypersensitivity [61]. Numerous human imaging studies further supported these findings [62,63]. In IBS patients, rectal distension, which induces pronounced pain/urge activation (hyperalgesia), is accompanied by increased activation of the ACC compared with controls [64,65].

Synaptic plasticity, such as long-term potentiation (LTP) and long-term depression (LTD), provides the neural basis for learning and memory [66,67]. Chronic pain has a vital emotional component, and cumulative evidence suggests that plastic changes in the ACC neurons play an important role in chronic pain [68,69]. Different stimulation protocols, including pairing training, the spike-excitatory postsynaptic potential (EPSP), and theta burst stimulation (TBS), induced LTP in ACC pyramidal neurons [70]. In addition, ACC pyramidal cells undergo rapid and prolonged depolarization in chronic pain model animals suggesting that ACC neurons might be associated with the synaptic mechanisms for pain [71]. The expression of immediate early genes, including c-fos, has been used as a marker for neuronal activation and is associated with the late phase of hippocampal LTP. Rodent studies with chronic inflammatory, neuropathic, and chronic visceral pain demonstrated increased c-fos in the ACC [72,73,74]. These results suggest that changes in synaptic transmission in chronic pain are linked to persistent LTP in the ACC. In summary, these studies indicate that the ACC generates a teaching signal for aversive experiences and mediates the affective/motivational aspects of pain.

### 3.2. The Amygdala

The amygdala (Figure 2), a limbic brain structure, critically mediates many aspects of emotional responses to various sensory stimuli, including pain [75,76,77]. Pain has a vital emotional component and is significantly associated with adverse effects [78]. The amygdala is made up of several functionally different nuclei, including the lateral (LA), basolateral (BLA), and central (CeA). Clinical neuroimaging data show that the BLA received sensory, including nociceptive, information from the thalamus and cortical areas such as the insula, ACC, and mPFC [79,80] and was activated in both acute and chronic pain [81]. BLA neurons project to the mPFC, and optogenetic manipulation of this circuit modulates both sensory and affective components of pain. The CeA receives nociceptive information from the spinal cord and the pontine parabrachial area (PB), forming a spino-parabrachial-amygdaloid pain pathway [82] and serves primary output functions for amygdala pain neurocircuitry. Both electrophysiological and anatomical studies suggest the involvement of the spino-parabrachio-amygdaloid pathway in the affective, emotional, and autonomic aspects of pain. Both thalamic and cortical sensory information integrate into the amygdala through connections with the LA-BLA nuclei, which then project to the CeA [76,83]. Electrophysiological studies demonstrated increased neuronal excitability in the CeA and BLA neurons [84] in acute arthritis [85,86,87] and the CeA in neuropathic pain [88,89], suggesting the enhanced excitatory synaptic transmission in the pain state of this neurocircuitry. A recent study uncovered the neural circuits that mediate the sensory and emotional aspects of pain by combining elegant techniques, such as viral tracing, optogenetics, fiber photometry, and electrophysiological recordings. The authors demonstrated that while optical activation of the IC→BLA pathway is responsible for regulating both the sensory and aversive components of pain, the activation of the MD→BLA pathway only contributes to the aversive-affective component of pain [90].

Several lines of evidence suggest the role of the amygdala in modulating visceral pain. Chemical activation of the CeA enhanced visceromotor responses to CRD in rats and caused the sensitization of spinal neurons [91]. In response to noxious stimuli, human IBS patients and rats demonstrated enhanced neuronal activity in the amygdala [92,93] and CeA [94,95], respectively. A recent study using colitis model rats investigated how visceral pain affects the electrophysiological properties of CeA neurons. Whole-cell recordings from these neurons showed enhanced synaptic transmission, as well as direct current injection, increased the frequency of evoked action potentials suggesting a critical role of the CeA in visceral pain. Altogether, these studies support the role of the amygdala in central pain processing.

### 3.3. Periaqueductal Gray (PAG)

The midbrain periaqueductal gray (PAG; Figure 2) plays a pivotal role in coordinating autonomic, motor, and pain-modulatory responses to relevant environmental stimuli [61,96,97]. The PAG receives selective inputs from the cortical and sub-cortical structures of the brain. Tracing studies from nonhuman primates and rodents revealed that the PAG receives connections from the mPFC [98], ACC [61], motor cortex [97], ventral insula, and amygdala of the brain [99]. Using diffusion tractography, similar functional connections were observed in human studies though not all connections correlate to animal studies [96,100]. Anterograde tracer injections in the PAG confirmed the existence of direct projections to the rostral ventrolateral medulla (RVM) [101,102]. Inputs from nociceptive SDH neurons reach the PAG through the spino-thalamic and spino-mesencephalic pathways [96,103]. In turn, the PAG exerts its pain modulatory effects via its descending projection to the RVM. These findings suggest that the PAG can exert its dual modulatory effect on pain perception through propagating nociceptive and analgesic stimuli [96] and serves as the hub for descending pain control [104,105].

Based on anatomical and functional observations, the PAG has been subdivided into dorsomedial (dmPAG), dorsolateral (dlPAG), lateral (lPAG), and ventrolateral (vlPAG) columns [96,106,107]. Earlier reports indicate that distinct PAG subregions interact with different pain-processing areas suggesting a plastic pain perception mechanism. Either electrical or chemical stimulation of the vlPAG suppresses nociceptive reflexes and elicits analgesia in somatic [108,109,110] and visceral pain animal models [111,112]. In addition, injections of low doses of morphine only in the vlPAG produce morphine-induced analgesia [113,114]. More recently, using ofMRI in mice it was found that the dl/lPAG induces active defensive behaviors against noxious stimuli [61]. Other studies demonstrated that optogenetically activating projections of layer 5 M1 neurons to the lPAG and vlPAG [97] and layer 5 mPFC neurons to the vlPAG [98] attenuated mechanical allodynia and thermal hyperalgesia in neuropathic mice. Dysregulation of the PAG functions is critical for developing and maintaining chronic pain states. Impaired functional connectivity of the PAG was observed in human patients and animal studies with neuropathic pain [115,116,117]. Similarly, enlarged grey matter volume [118,119], increased neuronal activity [120,121], and impaired neurotransmission [112,122,123] in the PAG were also observed in chronic visceral pain studies. Previous functional neuroimaging and electrical microstimulation studies have shown that somatic and visceral painful stimuli elicit diverse patterns of activity in the PAG [124,125,126]. A recent study used c-fos immunolabelling and extracellular microelectrode recording in urethane-anesthetized rats to investigate the colitis-induced changes in visceral pain-related neuronal properties of the PAG and its descending pathway. This study demonstrated diminished activation of the l/vlPAG by noxious CRD, and extracellular recording in the vlPAG revealed a colitis-elicited decrease in the proportion of CRD-excited neurons [127]. These studies suggest the existence of input-specific neural networks in the PAG that can undergo functional changes and contribute to the development of chronic abdominal pain. Altogether, these studies indicate the presence of input-specific neural networks in the PAG. Through both ascending and descending projections, the PAG can undergo functional changes and contribute to the development of chronic abdominal pain.

### 3.4. Locus Coeruleus (LC)

The brain nucleus locus coeruleus (LC) is located in the dorsolateral pons and is a significant source of norepinephrine (NE) in the brain (Figure 2). Retrograde tracing studies in rats demonstrated that ascending axons of noradrenergic LC neurons project to specific regions of the CNS including the thalamus, mPFC, ACC, hippocampus, hypothalamus, amygdala, and cerebellum [128,129], while descending axons project to the spinal dorsal horn (SDH) [130]. The LC releases NE to modulate nociception by altering the excitability of spinothalamic tract neurons [131,132,133]. These data suggest that LC and sub-coeruleus (SC) regions provide noradrenergic innervation to the forebrain, cerebellum, brainstem, and dorsal and ventral horns of the spinal cord [130,134,135]. Electrical stimulation of LC neurons produces antinociception [136,137,138], and intrathecal injection of noradrenergic antagonists [137,138] blocks these effects. Furthermore, chemical activation of the LC region relieves neuropathic pain [139], suggesting the analgesic role of LC. A recent study used chemogenetic, the Designer Receptors Exclusively Activated by Designer Drugs (DREADD), to activate the LC-SC pathway selectively and explore its effect on neuropathic pain in mice. Chemogenetic activation of the LC-SC pathway increased the release of NE in the SDH and reduced pain intensity in neuropathic mice [140]. This study and other studies also demonstrated that chemogenetic activation of astrocytes reduced neuroinflammation and restored cognitive deficits in VH rats [140,141]. Increased neuronal discharge in the LC was reported in response to visceral stimuli and is associated with cortical EEG activation [142,143]. LC receives multi-synaptic connections from other pelvic visceral organs, including the bladder, urethral sphincter, and kidney, through sympathetic and parasympathetic nerves [144,145,146], suggesting the role of the LC-NE system in pain perception arising from these organs [147,148]. Moreover, CRF microinjection into the rat LC/SC induced a prolonged lasting stimulation of colonic transit and bowel discharge [149] while inhibiting gastric secretion [150]. These observations further suggest the possibility that LC neurons can play a significant role in spinal nociceptive processing [140].

Acupuncture, electrical stimulation at ST 36 acupoint, has been widely used for treating VH in an animal model of FD [151,152] and IBS [153] in human patients. In an unpublished work from our group, we observed that bilateral electrical stimulation at ST36 reduced pain responses in male and female models of IBS rats. Experimental and neuroimaging studies in animals and humans have shown the involvement of many brain structures in the modulation of acupuncture-induced analgesia, including the RVM, PAG, LC, ACC, and the amygdala [154,155,156]. The involvement of these brain regions in pain processing is well documented. It has been shown that spinal α2-adrenoceptors play a crucial role in inhibitory descending pain control by noradrenergic projections from LC to the SDH [157]. A recent report indicated the involvement of spinal α2 receptors in acupuncture-induced analgesia. Altogether, these data support that LC may play a crucial role in the central pain processing mechanism.

### 3.5. The Hippocampus

In the previous sections, we already summarized that chronic pain reorganizes the structural and functional connections in both cortical and subcortical structures, including the mPFC [81,84], thalamus [158], amygdala [78,84], and ACC [159]. These brain areas are also associated with functions including navigation, learning and memory, fear, and emotional responses. Cognitive and emotional complications, such as anxiety and depression, and deficits in decision-making [160] and working memory [161] are associated with the FGIDs. The hippocampus (Figure 2) is a limbic brain area crucial for declarative memories in humans [162,163], encodes episodic and spatial memories in animals [66,164,165], and is involved in anxiety and depression [166,167]. Hippocampal pyramidal neurons fire action potentials when an animal visits a particular location of its environment. These cells are known as place cells and are thought to provide the basis for an internal representation of space, suggesting the role of the hippocampus in spatial learning and navigation [168,169]. Other studies further supported these findings by demonstrating direct or indirect connections between the hippocampus and the brain areas mentioned earlier [170,171,172,173,174,175,176,177,178,179,180,181]. However, the hippocampus has not yet been the focus of many studies on pain mechanisms.

A previous study suggested that the limbic forebrain structures, including the hippocampus, are responsible for modulating the aversive component of pain [182]. Electrophysiological studies from the hippocampus showed that nociceptive stimulus produces profound and prolonged depression in hippocampal CA1 population spikes [183]. Abnormal hippocampal activities, including short-term [184] and recognition memory deficits [185], are also observed in chronic pain model animals. Additionally, reduced hippocampal volume was observed in human patients with chronic pain [186]. The previous finding suggests that the spinothalamic and parabrachial pathways convey nociceptive information from the periphery to the hippocampus, while the septo-hippocampal pathway transfers these inputs directly from the spinal cord [186]. In summary, the hippocampus works with the ACC, the insula, the amygdala, and the PFC [187] to combine emotional and cognitive aspects of pain perception and may regulate chronic visceral pain in FGID patients [188,189].

## 4. Electrophysiological Recordings in FGID Model Animals

Converging evidence shows that to understand the pathophysiological basis of pain in FGIDs, it is necessary to precisely decode how the information is processed between the brain–gut axis and across different brain regions. Various methods have been applied to study the pain processing mechanism in the brain; however, the currently available techniques have yet to gain widespread clinical use. Neuroimaging techniques, such as fMRI and PET, measuring indirect neuronal activity, have been widely used to study pain processing. Although these methods have contributed significantly to understanding the pain mechanism, these techniques have a relatively poor temporal resolution. Consequently, a high temporal resolution method, such as electrophysiology (Figure 3), that can directly measure neuronal activity is needed to address pain processing dynamics. Neurons in the brain generate electrical pulses called action potential by integrating electrical inputs from thousands of other cells and performing fast computations that underlie animal behavior. Revealing the behavioral and cognitive relevance of the activity of neurons and their interactions requires monitoring these in the intact brain of behaving animals. Electrophysiology is a uniquely well-suited method combining high spatial and temporal resolutions (~millisecond), enabling us to record the brain’s natural language. Moreover, with ease of application, this user-friendly approach is an attractive tool for awake behavior preparation. In vivo electrophysiological recordings can be performed in anesthetized and awake animals while maintaining their normal behaviors. Recordings in anesthetized animals offer the opportunity to assess local field potentials (LFPs) and multiunit activity (MUA) with a minimum of artifacts at highly defined cortical synchronization states. However, the results also differ to some extent from what can be found in awake subjects. Extracellular microelectrodes are the most widely used tool for recording neuronal activity from the brain. Many variants of extracellular recording techniques are available. Largely these methods can be clustered into single-sited electrodes and multi-sited electrodes. Simultaneous large-scale neuronal population recordings can help to understand the coordinated activity underlying brain computations [190,191,192]. This process requires large dense arrays of recording sites, ideally compatible with freely moving rodents. Several silicone probes are available for chronic recordings [190,193,194]. These probes are designed with dense and extensive recording sites of up to 1024 electrodes to isolate individual neurons across large brain regions with minimal tissue damage. The probes have low noise, resistance to movement artifacts or other interference, and efficient data transmission, enabling us to record stable neuronal activity up to 8–10 weeks after implantation and ensuring low-cost scalable fabrication. The probes are designed to acquire electrophysiological measurements from a large number of neurons with unprecedented detail in freely moving animals. Their lightweight design, small footprint, and integrated fabrication enable the simultaneous monitoring of large populations of neurons from interconnected brain areas and the assessment of the relationship to behavior in normal and pathological conditions, making them ideal tools for recording brain signals from FGID animal models.

Here we summarized previous findings (Table 2) that used extracellular recording methods in animal models of FGIDs. The literature search for this review was conducted based on the Preferred Reporting Items for Systematic Reviews and Meta-Analysis (PRISMA) statement.

### Search Strategy

English databases, including PubMed (PM), and Web of Science core collection (WS), were searched for published articles without restricting the year of publication. The following keywords were used in the searching process: (1) visceral pain and extracellular recording in animals (PM; n = 70, WS; n = 14); (2) visceral pain and electrophysiological recordings in functional gastrointestinal disease (PM; n = 8, WS; n = 2), and (3) Electrophysiological recordings in functional gastrointestinal disease (PM; n = 73, WS; n = 10). Moreover, 18 articles were found in the citation search processing.

Inclusion and Exclusion criteria: Based on the focus of our review, we aimed to include studies that performed extracellular recordings from the brain in animal models of FGIDs. The following exclusion criteria were established: (1) studies conducted in human patients; (2) studies involving brain imaging techniques such as fMRI and PET; (3) non-empirical studies, such as editorial letters, conference proceedings, and literature reviews; (4) articles not written in English, or (5) studies performing intracellular recordings.

Selection process: From all search methods, 195 articles were identified. After removing duplicates, the titles and abstracts were retrieved and screened for eligibility according to the exclusion criteria. The number of reports was adjusted to 162, the full texts of the 34 articles were reviewed, and 9 papers matched the requirements and were included for reporting (Figure 4). Extracellular recording in the hippocampus is a well-established method; therefore, we reported a few studies that performed extracellular recordings from the hippocampus in rodents.

## 5. Insights from Large-Scale Neuronal Recording

Different brain regions are interconnected and must communicate to provide the basis for integrating information critical for learning, memory formation, perception, and the behavior of organisms. Cell assembly theory [201] suggests that changes in synaptic strength mediate this communication through synchronous cell activation. However, recent studies indicate that the interplay and coupling between neural oscillations, recorded by local field potentials (LFPs) [192] and electroencephalography (EEG) [202], provide a key mechanism for the communication between distributed networks in the brain. Neural oscillations are rhythmic or repetitive neural activity in the CNS; they range from slow [203,204] to medium ones, such as theta (4–8 Hz), alpha (8–13 Hz), fast ones, like beta (15–25 Hz), gamma (30–80 Hz), and, the ultra-fast ones (>100 Hz) [205,206,207]. Distinct neural oscillations are associated with different states of the brain; for example, theta oscillations are linked to memory encoding and retrieval [164], while gamma oscillations support cognition through the coupling of functional brain areas during associative learning [208]. Disruption of these brain oscillations is associated with pain perception in humans and model organisms [209,210]. Therefore, these ubiquitous rhythmic processes must be precisely and reliably quantified to understand their role in normal or abnormal brain function.

Brain oscillations are transient and emerge in bursts or packets. The principal approach to studying these oscillations from LFPs involves decomposing these signals into magnitude and phase information for each frequency and characterizing their changes over time (on a millisecond time scale). This approach is commonly used for hunting oscillation bursts in neural data and is known as time-frequency analysis [211,212]. There are many approaches for capturing different aspects of magnitude and phase relationships of neural signals, such as the short-term Fourier transform (STFT) [213], the discrete or continuous wavelet transform, the Hilbert transform, and matching pursuits [214]. Time-frequency analyses of LFPs quantify the power and phase synchrony of specific frequency bands across time and space and provide information about neural synchrony not apparent in the ongoing neural signals.

Communication between selective brain structures is a crucial component of the neural processing underlying cognition [215,216]. Electrophysiological studies suggest that synchronous neural oscillations play a vital role in the flexible routing of information flow in the brain; this is proved by changes in the coherence of LFPs [192,217] and cross-correlated unit activity. In particular, theta rhythms are prominent oscillations recorded in the hippocampus during various locomotor activities such as voluntary, preparatory, orienting, or exploratory and during REM sleep [218]. Theta waves are absent when animals are immobile, but short epochs of theta trains can be elicited by noxious conditioned stimuli [218]. Theta oscillations are present in cortical and subcortical structures, including the entorhinal cortex, perirhinal cortex, ACC, and amygdala [218]. However, these cortical structures are not capable of generating theta activity on their own. Earlier studies demonstrated that theta rhythm is important in forming and retrieving episodic and spatial memory [219]. Despite extensive research on theta oscillations in cognition and memory, clinical studies suggest that chronic pain is associated with abnormal theta oscillatory activities [197,220,221,222]. Power spectral analysis of the resting EEG of neurogenic pain patients exhibited higher resting-EEG power at 2–25 Hz with the maximum difference in the theta frequency band [221]. In addition, a higher baseline level of delta and theta EEG oscillations was observed in patients with visceral [223] and somatic pain syndromes [224]. Alternations of these oscillatory activities in chronic pain states indicate disruption of local or long-range communication between functionally specialized neuronal assemblies.

## 6. Concluding Remarks and Future Directions

Increasing evidence strongly indicates that the crosstalk between the gut and brain is important for understanding visceral pain pathophysiology in FGIDs. However, the mechanisms by which these regions communicate are still in their early infancy. The brain receives noxious pain stimuli through anatomically and functionally distinct medial and lateral pain pathways. The lateral (lateral thalamus→S1) pathway encodes sensory–discriminative attribution of nociception. The medial (MD→ACC) pain system contributes to the motivational–affective component of pain arising from deep somatic and visceral structures. In chronic pain states, both pain pathways are known to be disrupted. Human and animal studies suggest that disruption of brain oscillatory activity is associated with neurological and GI disorders. However, the precise mechanisms underlying the disturbances of the rhythm coherence in FGIDs still need to be better understood.

The role of theta oscillations in cognition is widely accepted, and emerging evidence suggests pain specific ongoing activity in the theta band. Therefore, theta oscillation dynamics can serve as a basis for decoding pain sensation. Future works in this field are beneficial for identifying the oscillatory basis of pain perception in FGIDs and provide a deeper understanding of the nociceptive systems in these disorders. Electrophysiological methods such as large-scale neuronal population recordings could be suitable for studying pain mechanisms because of their relatively low cost, high temporal resolution, and ease of use compared to other neuroimaging techniques. In clinical practice, this also offers exciting prospects for investigating pathological mechanisms of visceral pain in FGIDs, thus promoting the development of a rational therapeutic strategy.

## Figures and Tables

**Figure 1 diagnostics-13-00627-f001:**
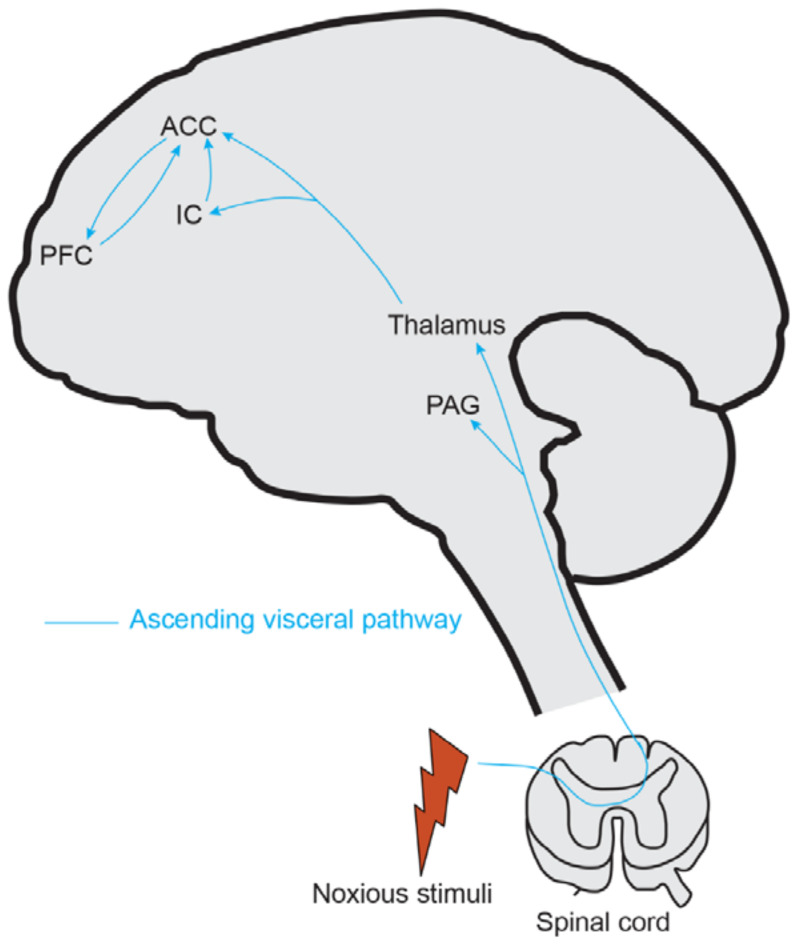
Ascending pain modulation. The ACC and IC receive nociceptive signals from the SDH through the thalamus. ACC: Anterior cingulate cortex, IC: Insular cortex.

**Figure 2 diagnostics-13-00627-f002:**
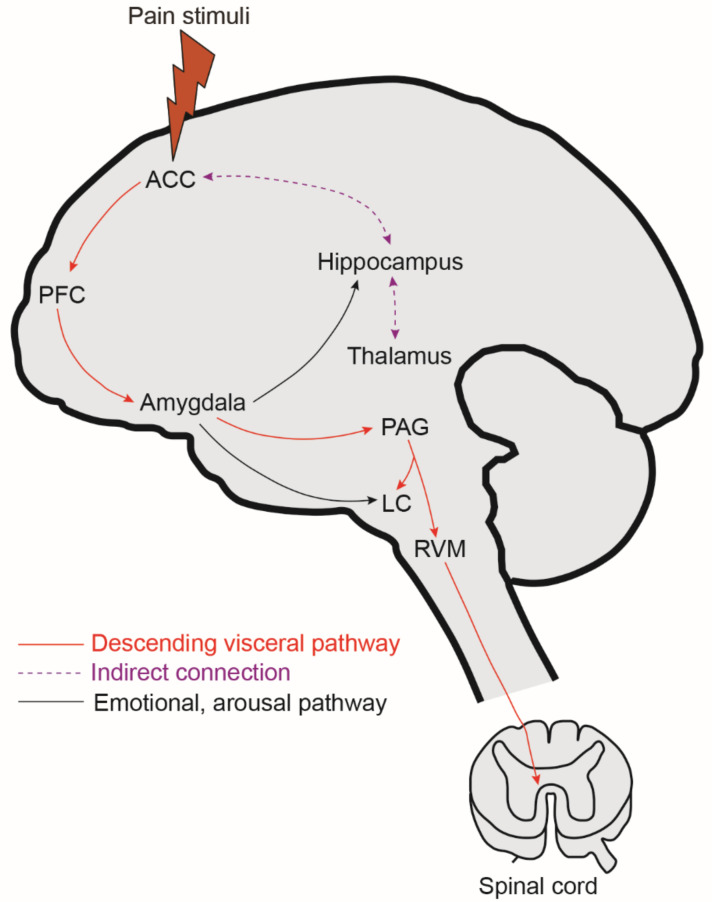
Descending pain modulation. The pain signals from the cortical brain regions (ACC, mPFC) reach the PAG and LC. PAG and LC exert their pain modulatory effects via their descending projection to the RVM and spinal cord, respectively. ACC: Anterior cingulate cortex, PFC: Prefrontal cortex, PAG: Periaqueductal gray, LC: Locus coeruleus, RVM: Rostral ventrolateral medulla.

**Figure 3 diagnostics-13-00627-f003:**
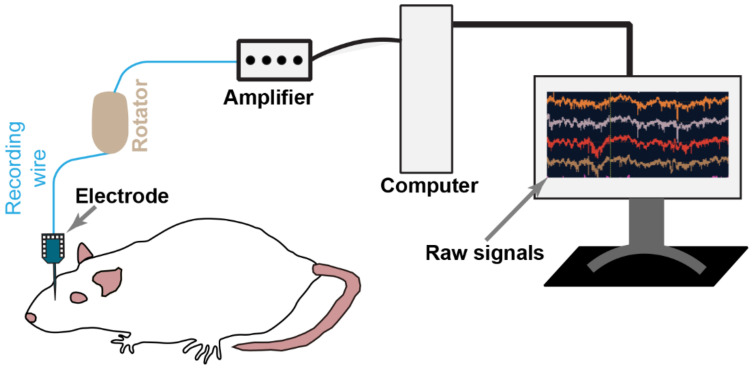
Flowchart of the chronic extracellular recordings in model animals.

**Figure 4 diagnostics-13-00627-f004:**
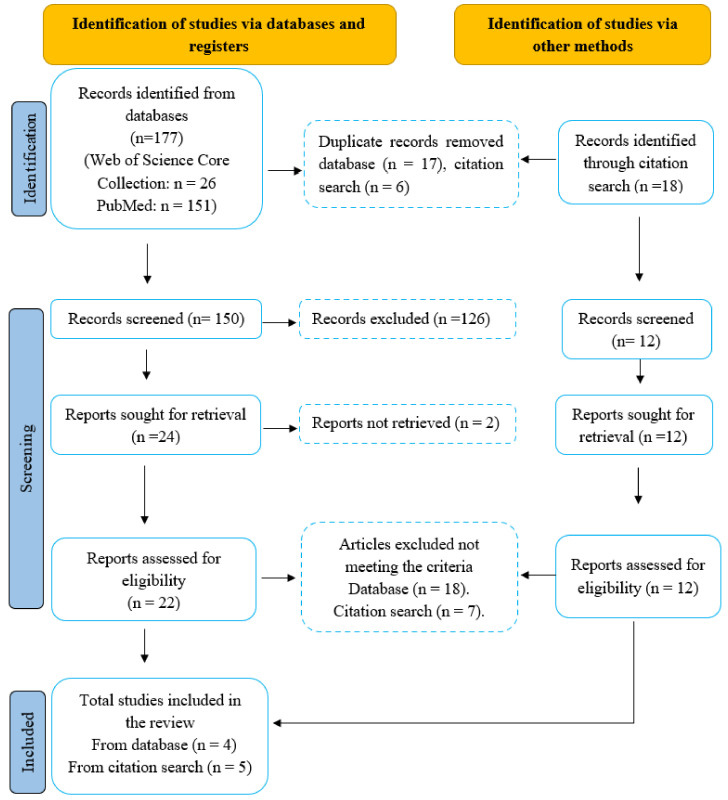
PRISMA flow diagram. Note: Following the screening and reviewing process guidelines based on Page et al. [195].

**Table 1 diagnostics-13-00627-t001:** Comparisons of models to assess mechanisms of VH in animal models.

Animal Model	Species, Sex, and Intervention	Findings	References
VH	Rat, Male, WAS	These animals had higher corticosterone and increased mass of the adrenal gland.	Myers et al., 2012 [33]
VH	Rat, Male, WAS	CB1 receptor expression decreased, and TRPV1 receptor expression increased. DNA methylation and histone acetylation of genes responsible for visceral pain sensation also increased.	Hong et al., 2011 [41] Hong et al., 2015 [42]
VH	Rat, Male, AA	Elevated NE concentration in blood plasma, no change in the β2AR expression	Zhu et al., 2015 [43]
VH	Rat, Male, AA, NMD, AMS	Upregulation of the P2Y_1_ and P2X_3_ receptors was observed in the colonic DRG, the expression of PKC protein in the spinal cord, and the enhancement of ATP-induced current density.	Zhao et al., 2020 [44] Xu et al. [45] Hu et al., 2020 [46]
Colitis	Rat, Male, TNBS	This study reported increased colon weight, IL-6, and IL-10 production, higher fibrosis score, and increased anxiety-related behavior. No difference was observed in intestinal permeability, colonic TNF-α, MPO activity, and TRPV1 expression.	Salameh et al., 2019 [47]
VH	Rat, Male, NCI, MO	Recording from viscero-sensitive SDH neurons showed higher activity. Interestingly this study reported no identifiable pathological changes in the colon.	Al-Chaer et al., 2000 [20]
VH	Rat, Male, CRF	CRF injection significantly increased NE and dopamine release in the CeA while not affecting the serotonin level.	Su et al., 2015 [27]
Colitis	Rat, Male, TNBS	Shorter colon length, increased spleen weight, decreased IL-10, but increased IL-6, IL-17A, MPO, and TNF-α production.	Gou et al., 2019 [24]
Colitis	Mouse, Male, DSS	These mice had diarrhea, rectal bleeding, weight loss, erosions, and inflammatory changes, including crypt abscesses in the large intestine, prominent regenerations of the colonic mucosa, shortening of the large intestine, formation of lymphoid follicles, increased intestinal microflora, *Bacteroides distasonis,* and *Clostridium* spp.	Okayasu et al., 1990 [23]
VH	Mouse, Male, NMD	Higher neuronal discharge in the CL and ACC in response to CRD; increased NMDA receptors and overactive CaMKIIα in the ACC. Optogenetic inhibition or activation of CL either suppressed or enhanced the ACC neural activity and pain sensitivity, respectively.	Xu et al., 2022 [48]

Abbreviations: Acetic acid (AA), Adenosine triphosphate (ATP), Adult multiple stress (AMS), Anterior cingulate cortex (ACC), Calcium/calmodulin-dependent protein kinase type II subunit alpha (CAMKIIα), Cannabinoid receptor 1 (CB1), Claustrum (CL), Central nucleus of the amygdala (CeA), Colorectal distension (CRD), Corticotrophin releasing factor (CRF), Dextran sodium sulfate (DSS), Dorsal root ganglion (DRG), Interleukin (IL), Mustard oil (MO), Myeloperoxidase (MPO), Neonatal colonic irritation (NCI), Neonatal maternal separation (NMD), Noradrenaline (NE), N-methyl-D-aspartate (NMDA), Protein kinase C (PKC), Purinergic receptor (P2Y1 and P2X3), The β2-adrenergic receptor (β2AR), Tumor necrosis factor-α (TNF-α), Transient receptor potential vanilloid subfamily, member 1 (TRPV1), Visceral hypersensitivity (VH), Water avoidance stress (WAS), and 2,4,6-Trinitrobenzene sulfonic acid (TNBS).

**Table 2 diagnostics-13-00627-t002:** Summary of the reviewed literature.

Article	Model	Species and Animal Number	Recording Site and Techniques	Main Findings
Lyubashina et al., 2022 [127]	Colitis	Rat Control (n = 26) Colitis (n = 26)	PAG Tungsten microelectrode	PAG neuronal recording revealed a colitis-elicited decrease in the proportion of CRD-excited neurons and an increase in unresponsive cells, suggesting the role of PAG in ascending and descending visceral nociception control.
Gao et al., 2006 [196]	VH	Rat Control (n = 15) VH (n = 22)	ACC Glass microelectrode	ACC neurons had an increased firing rate compared to control rats in response to CRD, suggesting the sensitization of the ACC in VH rats.
Cao et al., 2016 [197]	VH	Rat Control (n = 6) VH (n = 6)	BLA, ACC Multi-channels electrode	ACC neurons in VH rats had disrupted spikes phase-locking to theta oscillations. Moreover, they found desynchronized theta activities between BLA and ACC.
Xu et al., 2022 [140]	VH	Mice Control (n = 8) NMD (n = 28)	Claustrum, ACC Multi-channel electrode	Discharge of action potentials increased in ACC and Claustrum in response to CRD.
Hasan et al., 2023 [141]	VH	Rat (n = 10)	BLA, ACC Multi-channels electrode	VH caused hypomyelination in the ACC neurons, decision-making deficits, and impaired ACC-BLA synchrony in the theta band. Chemogenetic activation of astrocytes recovered VH-induced desynchronization.
Ji et al., 2015 [198]	Colitis	Rat (n = 12)	CeA Glass insulated carbon filament electrodes	CeA neurons showed increased firing in response to graded CRD, suggesting the presence of vis-cero-nociceptive neurons in the CeA.
O’Keefe et al., 1971 [169]	Wild type	Rat (n = 23)	Hippocampus Silicone probe	This is the first study demonstrating place-cell in the hippocampus, suggesting that the hippocampus provides a reference map to the rest of the brain.
Diba et al., 2007 [199]	Wild type	Rat (n = 3)	Hippocampus Silicone probe	This study demonstrated that hippocampal place-cell sequences are fired in forward and reverse order and replayed similarly during subsequent sleep.
Vandecasteele et al., 2012 [200]	Wild type	Rat	Hippocampus Silicone probe	This methodological study provides detailed steps for recording multiple single neurons and LFPs by movable silicone probes in behaving animals.

## Data Availability

Not applicable.

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
