# Peer review of "Electrophysiology as a Tool to Decipher the Network Mechanism of Visceral Pain in Functional Gastrointestinal Disorders"

_diagnostics, 2023, doi:10.3390/diagnostics13040627_

Round 1

Reviewer 1 Report

This review paper, titled by "Electrophysiology as a tool to decipher the network mechanism of visceral pain in functional gastrointestinal disorders", is very interesting in the field of visceral pain from Gut. It reviewed most of investigations in correlation between Brain and visceral hypersensitivity,including method,mechanism. 

There are some comments to revise as following:

1, Is any different in the mechanism of visceral hypesentivity models,such as stress,and colitis? And changes in the different part of Gut?

2,Please add a table to give a summary in the effect of different stimulation,or different parameters.

3,Please change the  first figure "colon" to "Gut".

Author Response

Dear reviewer,

Thank you very much for your comments concerning our manuscript entitled “Electrophysiology as a tool to decipher the network mechanism of visceral pain in functional gastrointestinal disorders”. These comments are valuable and helpful for revising and improving our manuscript. We have studied these comments carefully and have made corrections which we hope to meet with approval. Modified portions are marked with the “Track Changes” option in the paper. The significant modifications in the article and the responses to the reviewer’s comments are as follows:

1,  Is any different in the mechanism of visceral hypersensitivity models, such as stress, and colitis? And changes in the different part of Gut?

Response: Thank you for your fascinating question. It is worth comparing the difference among different animal models; however, to the best of our knowledge, the data is limited as all animal models share the standard mechanisms for inducing VH though some differences still exist. In the revised version of the manuscript, we have summarized these observations in Table 1.

Table 1. Comparisons of models to assess mechanisms of VH in animal models.

Animal model

Species, sex, and intervention

Findings

References

VH

Rat, Male, WAS

These animals had higher corticosterone and increased mass of the adrenal gland.

Myers et al., 2012 [31]

VH

Rat, Male, WAS

CB1 receptor expression decreased, and TRPV1 receptor expression increased. DNA methylation and histone acetylation of genes responsible for visceral pain sensation also increased.

Hong et al., 2011 [39]

Hong et al., 2015 [40]

VH

Rat, Male, AA

Elevated NE concentration in blood plasma, no change in the β2AR expression

Zhu et al., 2015 [41]

VH

Rat, Male, AA, NMD, AMS

Upregulation of the P2Y1 and P2X3 receptors was observed in the colonic DRG, the expression of PKC protein in the spinal cord, and the enhancement of ATP-induced current density.

Zhao et al., 2020 [42]

Xu et al., [43]

Hu et al., 2020 [44]

Colitis

Rat, Male, TNBS

This study reported increased colon weight, IL-6, and IL-10 production, higher fibrosis score, and increased anxiety-related behavior. No difference was observed in intestinal permeability, colonic TNF-α, MPO activity, and TRPV1 expression.

Salameh et al., 2019 [45]

VH

Rat, Male, NCI, MO

Recording from viscero-sensitive SDH neurons showed higher activity. Interestingly this study reported no identifiable pathological changes in the colon.

Al-Chaer et al., 2000 [19]

VH

Rat, Male, CRF

CRF injection significantly increased NE and Dopamine release in the CeA while not affecting the serotonin level.

Su et al., 2015 [25]

Colitis

Rat, Male, TNBS

Shorter colon length, increased spleen weight, decreased IL-10, but increased IL-6, IL-17A, MPO, and TNF-α production.

Gou et al., 2019 [21]

Colitis

Mouse, Male, DSS

These mice had diarrhea, rectal bleeding, weight loss, erosions, and inflammatory changes, including crypt abscesses in the large intestine, prominent regenerations of the colonic mucosa, shortening of the large intestine, formation of lymphoid follicles, increased intestinal microflora, Bacteroides distasonis, and Clostridium spp.

Okayasu et al., 1990 [22]

VH

Mouse, Male, NMD

Higher neuronal discharge in the CL and ACC in response to CRD; increased NMDA receptors and overactive CaMKIIα in the ACC. Optogenetic inhibition or activation of CL either suppressed or enhanced the ACC neural activity and pain sensitivity, respectively.

Xu et al., 2022 [46]

2, Please add a table to give a summary in the effect of different stimulation, or different parameters.

Response: Thank you very much for your comments. However, we did not review electrical stimulation in this article; therefore, no information can be provided.

3,Please change the  first figure "colon" to "Gut".

Response: Thank you very much for your suggestions. We have modified the figure according to your proposal.

We appreciate your warm work earnestly and hope the correction will be approved.

Thank you very much for your comments and suggestions.

Reviewer 2 Report

This manuscript is very well designed and contribute to new approach to related data.

Author Response

Dear reviewer,

Thank you very much for reviewing our manuscript entitled “Electrophysiology as a tool to decipher the network mechanism of visceral pain in functional gastrointestinal disorders”. Your comments are valuable and helpful for revising and improving our manuscript. Once again, thank you very much.

We appreciate your warm work earnestly and hope the correction will be approved.

Thank you very much for your comments and suggestions.
